# Tweets on the Go: Gender Differences in Transport Perception and Its Discussion on Social Media

**Paula Vasquez-Henriquez [1,\*], Eduardo Graells-Garrido [1,2]** and **Diego Caro [3]**

1   Data Science Institute, Universidad del Desarrollo, Las Condes, Santiago 7610658, Chile;
    eduardo.graells@bsc.es
2   Barcelona Supercomputing Center (BSC), 08034 Barcelona, Spain
3   Departamento de Ingeniería Informática, Universidad de Santiago de Chile, Estación Central,
    Santiago 9170002, Chile; diego.caro.a@usach.cl
\*   Correspondence: pvasquezh@udd.cl

**Abstract:** People often base their mobility decisions on subjective aspects of travel experience, such as time perception, space usage, and safety. It is well recognized that different groups within a population will react differently to the same trip, however, current data collection methods might not consider the multi dimensional aspects of travel perception, which could lead to overlooking the needs of large population groups. In this paper, we propose to measure several aspects of the travel experience from the social media platform Twitter, with a focus on differences with respect to gender. We analyzed more than 400,000 tweets from 100,000 users about transportation from Santiago, Chile. Our main findings show that both genders express themselves differently, as women write about their emotions regarding travel (both, positive and negative feelings), that men express themselves using slang, making it difficult to interpret emotion. The strongest difference is related to harassment, not only on transportation, but also on the public space. Since these aspects are usually omitted from travel surveys, our work provides evidence on how Twitter allows the measurement of aspects of the transportation system in a city that have been studied in qualitative terms, complementing surveys with emotional and safety aspects that are as relevant as those traditionally measured.

**Keywords:** gender differences; social media; topic modeling; sentiment analysis; transport perception

## 1. Introduction

Transport experience varies from different groups within a population [1]. Men and women have different travel characteristics [2–4], place different degrees of importance on mobility-decision factors (e.g., time, cost, accessibility, frequency, safety), and move differently across the city depending on their daily activities [5–8]. This difference is critical for the sustainability of cities, as stated in one of the Sustainable Development Goals by the United Nations, which aims to "provide access to safe, affordable, accessible, and sustainable transport systems for *all*, [...] with special attention to the needs of those in vulnerable situations, *women* [...]" (emphasis ours). (https://www.undp.org/content/undp/en/home/sustainable-development-goals/goal-11-sustainable-cities-and-communities.html).

Advancing toward reaching this goal is not an easy task. One of the main difficulties is quantifying the differences in population groups. It is vital for transport planners to take into consideration the trends and behaviors of different segments of the population and understand their context (conditions, opportunities, and constraints) [9]. Traditional methods to gather information on travel experience often observe users as one "average user" group, without considering the heterogeneity among groups of people. For instance, surveys are the most common tool to measure the travel

experience [10,11]. Typically, their design is targeted to optimize quantitative characteristics such as costs for the system, while maintaining or improving quality of service, mainly, time and frequency. Even though they offer rich information, surveys have limitations. Surveys are not created to consider the multi-dimensional aspects of perception when people travel [12], leading to conclusions assuming that the travel experience is the same for both genders [13]. Reportedly, that is not the case. For instance, women experience higher levels of violence and harassment in transportation, which leads them to respond in ways that affect their travel experience and quality of life [14]. Many of these incidents are under-reported to the authorities, thus lowering the probability of this problem being accurately reflected in urban and transport planning [15]. The reasons include entrenched gendered power hierarchies and the lack of robust information about women's needs.

The ubiquity of mobile phones has allowed people to express their opinions and daily experiences on social media platforms, creating a vast source of unstructured data on implicit user satisfaction. We propose that social media data may complement surveys and other methods. In particular, in this study, we analyze gender differences in transport perception through the analysis of posts on Twitter. We hypothesize that, if gender influences travel experience in any way, this should be reflected by a difference in the linguistic components of the texts published in social media. Measuring the perception and its potential gender differences is a challenging task due to the unstructured data and the informal nature of social media.

The research question that drives our study is: How can we measure the gender differences in transport perception presented in social media? To answer this question, we develop a three-step process to analyze transport-related opinions expressed on social media platforms. First, we infer gender for user profiles, which is not directly available on micro-blogging sites. Second, we extract the latent structure from the discussion using a topic modeling approach [16,17], which allows to separate the personal experience (e.g., feelings about a specific trip by bus) from the general discussion (e.g., opinions on public transport policy). Finally, we quantify perception and its differences with respect to gender with an analysis grounded on psycho-linguistic theory, which allows us to study the affective and relative aspects of transportation revealed by users.

With this methodology, we study the Twitter transport perception in Santiago (Chile) during 2017–2018. We chose Chile as it has been the most connected country in Latin America since 2017 [18], and Santiago as it is the largest city in Chile, with a population of 7 million people. Santiago also poses a wide offer of transportation modes, including metro, bus, taxi, shared taxi, public bikes, and several ride-share apps for cars, scooters, and bikes. We chose Twitter as it is one of the most accessed applications from mobile phones in Santiago [19].

The main contributions of this work are two-fold. The first contribution is a methodology to analyze the discussion about transportation in social media, with a focus on gender differences. This contribution could help data scientists to extract new applied insights from already available data. The second contribution is a case study of measured gender differences, using Twitter data from a big city. These contributions could help urban planners to widen their understanding of how different groups of people experience one of the most recurrent daily activities, transportation, using a promising data source in planning for sustainability: social media [20].

This paper is structured as follows. Section 2 presents the related work about social media analysis and transportation experience measurement. Section 3 introduces the data and the proposed methodology for measuring gender differences in transport perception using social media data. Section 4 shows the results for a case study in Santiago, Chile. Section 5 presents the implications and main limitations of this work, as well as future work. The conclusions are given in Section 6.

## 2. Related Work

There are two main areas of research related to this work: Social Media Analysis and Transportation Experience Measurement. Here we discuss both in relation to the contributions of this work.

## 2.1. Social Media Analysis

In the field of Social Media Analysis (SMA), we identify three sub-areas relevant for this work: demographic analysis, sentiment analysis, and topic modeling.

Demographic analysis is essential, as social media platforms are not representative of the entire population in terms of age, gender, and socio-economic status [21]. Data obtained from this type of source cannot be generalized, although it is possible to de-bias the results from the analysis by using census data and specific methods [22]. In our work, we put the focus on gender, a specific demographic attribute of user profiles. Since this attribute is not always explicit, researchers need to infer them using machine learning methods [23]. Then, we analyze each group in separate, meaning that even if there are imbalances in the analyzed population groups, the insights regarding each gender should not be affected by this imbalance.

Sentiment analysis is the measurement of polarity in text, mainly between negative, neutral, and positive [24]. Methods to extract polarity commonly start with a subjectivity lexicon of words that are associated with positive and negative feelings [25], which are then used to train a classifier [26]. Note, however, that the sentiment in a text may go beyond being positive or negative. Other approaches take into account the multidimensional aspects of speech. To the extent of our knowledge, the most common is Linguistic Inquiry and Word Count (LIWC) [27], which includes multiple emotions and other topical dimensions of expression such as psychological and cognitive components. LIWC usage with social media data includes predicting postpartum changes in emotion [28], identifying health issues [29,30], and predicting political sentiment [31]. Our work is based on LIWC as a proxy of perception in transport (instead of other issues or themes), as it is available in Spanish (the language we analyze) and it has a proven track record in analyzing perception in different issues.

Topic modeling is a family of approaches to extract thematic clusters from unstructured collections of text. In the case of social media, this refers to inferring the several themes, or topics, that people discuss. The most well-known methods are Latent Dirichlet Allocation (LDA) [32] and Non-Negative Matrix Factorization (NMF) [33]. Both methods infer latent topics in text, where words and documents are associated with these topics in a quantifiable way. LDA follows a generative approach, aiming at finding the most likely word and topic distribution given the text under analysis. NMF decomposes documents (or arbitrary objects represented as a non-negative matrix) into the sum of its latent parts. When choosing one method over the other, the task to be solved needs to be considered as well as the type of text. In short texts (which are commonly found in Twitter), NMF and its variations tend to exhibit topics of better quality [34]. In similar contexts to ours, NMF has been used to characterize urban areas according to their tweets [35], to infer political leaning of words and user profiles [36], and to separate the several topics related to depression discussion [37], among others. A potential limitation in this area is that latent topics are not always interpretable. To guide the topic extraction toward a predefined set of human-interpretable topics, a semi-supervised method can be used [38]. In topic modeling, these methods require modelers to label a fraction of the dataset, where labels may be applied to documents (such as Labeled-LDA [39]) or either to documents or keywords, such as in Topic-Supervised NMF (TS-NMF) [16]. Here we use TS-NMF due to its direct formulation, as well as its precedents in being used jointly with LIWC to characterize perception in at least two thematic discussions: migration [40] and mode of transportation usage [41].

## 2.2. Transportation Experience Measurement

The concept of transport perception can be defined as the social value of the transport experience, while transport service quality can be considered as the sum of the users perceptions of that experience. Transport perception is generally estimated focusing on quality related factors such as accessibility, cost, frequency, reliability, comfort, and safety or through the measurement of subjective well-being. Although the mobility decisions of users of public transportation are assumed to maximize utility and satisfaction [42], transport perception has been proven to directly influence future user behaviour [43].

Some studies involving quality related factors include the creation and application of surveys which explicitly ask to participants to rank factor importances [44–49], and the generation of models to measure the the overall satisfaction considering factor contributions [50–55]. Many studies have attempted to identify peculiar characteristics of subpopulation groups on travel perception and behavior. These include defining traveler profiles based on the disaggregation of behavioral, socio-economic and psychosocial variables that map common attitudes, perceptions, and experiences [56]; the differentiation of different periods of the day (holiday or non-holiday) and visitor profile (tourist or non-tourist) [57]; the usage of mode of transportation also shapes experience, as socio-economic characteristics alone do not explain travel patterns [58]; women place more importance on factors such as road safety and punctuality regarding perception of quality for interurban bus services [59]; and finally, there is a potential of mutual influence between attitudes and behavior over time [60].

The aforementioned works imply that analyzing the experience and its perception is no easy task. However, methods to measure perception are arguably standardized. The study of transport perception through the measurement of subjective well-being is based on Satisfaction with Travel Scale (STS) [61]. This scale considers both cognitive judgement (self-reported ratings) and affective judgement of satisfaction (duration and intensity of affects during a given time span) to measure perception. A study on travel and time use on subjective well-being proved accessibility in transportation as a key factor towards improving the quality of life of women [62]. This further highlights how transportation analysis should consider the particular gender barriers in the access to the city [63].

The role of social media in transport analysis has rapidly grown over the last years, allowing the ability to obtain information regarding trips [64] and activities [65], while highlighting the benefits of using these unstructured data sources [66]. There are a few approaches to understand transport perception using this data source: in our literature search, we found a descriptive analysis of public transport perception from tweets in the city of Chicago [67], the monitoring of malfunctions in public transport in Madrid [68], the analysis of satisfaction with public transport in Santiago [69], and the public transport opinion (in terms of polarity, from negative to positive) in Nanjing [70]. Our work can be seen as a deeper analysis of the travel experience than these previous works, which are primarily focused on the polarity of messages, without considering gender differences nor more nuanced perception categories. This awareness of travel experience differences could allow transportation planners to consider a wider range of needs and dynamics into their work, beyond the needs of the business, complementing traditional data sources with fine-grained data.

## 3. Data and Methodology

In this section, we describe the dataset and methodology used to answer our research question. The context is the discussion about transportation from the city of Santiago, Chile. This discussion is contained in a collection of messages from the micro-blogging platform Twitter.

### 3.1. Social Media Dataset

Twitter is a micro-blogging platform that allows people to publish and exchange posts up to 280 characters, called tweets. Tweets may contain mentions to other users, hashtags to indicate topics of the post, website addresses, emoji, etc. Given that it is one of the most accessed applications from mobile phones in Santiago [19], we expect that people will frequently report their daily experiences, including transportation. We collected tweets related to transportation from the years 2017 and 2018. We used a manually crafted dictionary of keywords to query the Twitter streaming Application Programming Interface (API). The list contained transportation-related words, hashtags, mentions of transportation accounts, URLs, and transportation emojis. Samples of these query terms can be found in Table 1.

**Table 1.** Samples of keywords used to crawl transportation tweets. In parentheses, translations or explanations.

| Sample Keywords | | |
|---|---|---|
| bus stop (*paradero*) | easy taxi (ride-hailing application) | gas (*bencina*) |
| ride bicycle (*andar en bicicleta*) | panne (to run out of gas) | fare meter (*taxímetro*) |
| @uber_chile (ride-hailing application) | @mfc_oficial (bike organization) | 🚌 |
| @cabify (ride-hailing application) | train (*tren*) | 🚙 |
| bus-only lane (*corredor de buses*) | @metbus (bus operator) | #baquedano (*metro station*) |
| waze (GPS, route application for cars) | bike (*bici*) | subway station (*estación de metro*) |
| @autop_central (highway operator) | wagon (*vagón*) | 🚒 |
| highway (*autopista*) | 🏍 | #electricalbuses (#*buseselectricos*) |
| #publictransport (#*transportepublico*) | bus | 🚕 |

In total, we collected 443,000 tweets from 112,000 users living in the Santiago Metropolitan area. A manual inspection of these tweets showed several themes within transportation, e.g., complaints mentioning the subway service provider, announcements of new bicycle paths, and general complaints about the size of the children safety chairs in cars (see Table 2 for example tweets).

**Table 2.** Sample tweets from our dataset. The original tweets are in Spanish, we have translated them here for clarity.

| Quote # | Tweet |
|---|---|
| 1 | @metrodesantiago An escalator breaks and the whole line is down? Over an hour from Macul to Tobalaba 😡😡😡 |
| 2 | Recognize and locate the new bike lanes that @Muni_Stgo is implementing. Plan your mobility this #march |
| 3 | In conclusion, if you have 3 children under the age of 9, you must change your car for a 3 row van!! #semana24 |
| 4 | @metrodesantiago the smell of tobalaba station is very toxic, please put masks on the poor people who work there all day 😊 |
| 5 | There are way too many uber drivers who don't know Santiago 🤦 |
| 6 | @sebastianpinera @metrodesantiago @GloriaHutt @MTTChile @J_LDominguez @louisdegrange And the humiliation of subway drivers? Any public apology? Very unfortunate comment @sebastianpinera minimizing the work of the drivers of @metrodesantiago … Do you do your job well? To a part of the population, not less, it also seems little. Remember it. |
| 7 | @mbachelet L3 I look forward to it 🤗 |
| 8 | @Transantiago thanks!!! :) |
| 9 | @UOCT_RM How are the traffic lights now?, I have to go to the O'Higgins park from Quinta normal |
| 10 | @Transantiago bus line 223 took 40 min to get to pb114 stop 😡 this always happens in the evenings! terrible |
| 11 | All traffic returned to Santiago of Chile. |
| 12 | Las Condes will saction street harassment @TroncoTorrealba I hope Vitacura will also implement this! I say it because for months I walked scared across Bicentenary Park (part of my travel route) where I was harassed by a man who parked cars. |
| 13 | @valderrax Do you think street harassment does not exist? Eg I was riding a bicycle and a guy told me: I would like to be the saddle to have my face in your ass. Is that an unfortunate phrase? and isn't it harassment? Doesn't it affect my freedom? |
| 14 | 20 min between my office's door to the exit of National Stadium station. I LOVE YOU LINE 6 OF @metrodesantiago 😍 |
| 15 | Again buses 425 won't stop after 00:00 hrs. What's going on? Don't you oversee? I'm angry. @Transantiago @DTPMet @AlsaciaExpress |

The weekly publishing rate of those tweets is not uniform in time; on the contrary, people react to certain events. The periods with most volume have identifiable events related to transportation, but not necessarily to the transportation experience. Topics include public transport service shutdowns, malfunctions, and massive power outages, among others (see Figure 1). The daily frequency also

changes from day to day, as more tweets are published during business days than on weekends (see Figure 2).

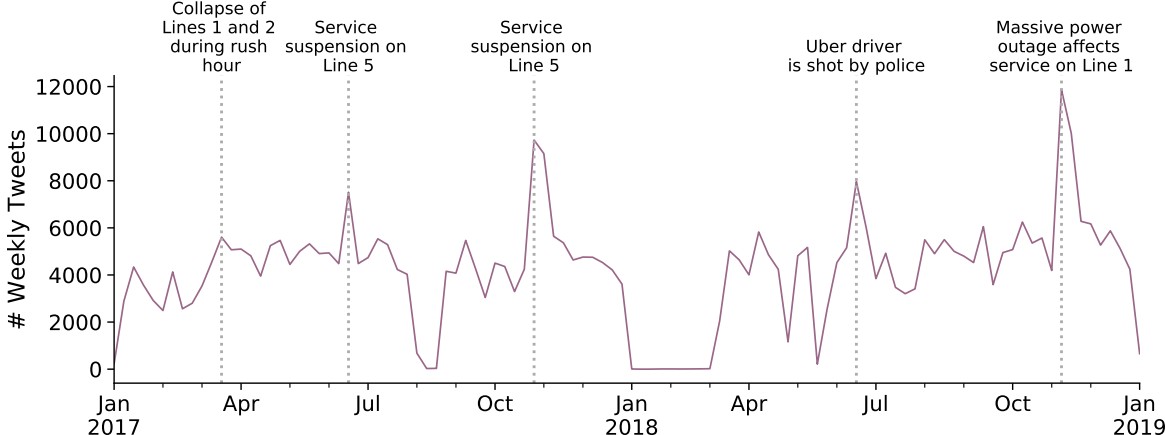

**Figure 1.** Transportation tweeting frequency during the years 2017 and 2018. Periods with zero tweets correspond to shutdowns in the tweet collection program.

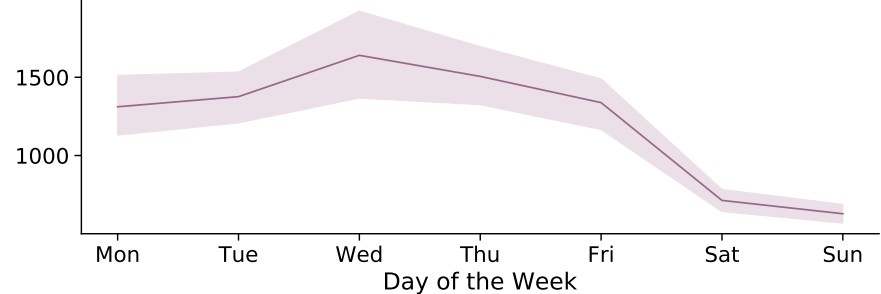

**Figure 2.** Average number of tweets per day of the week with a 95% confidence interval.

The hourly frequency distribution exhibits morning and evening peaks related to commuting times in Santiago (see Figure 3). It does not resemble the general usage of the application in the city [19]. For instance, during lunch hours, people tweet less about transportation.

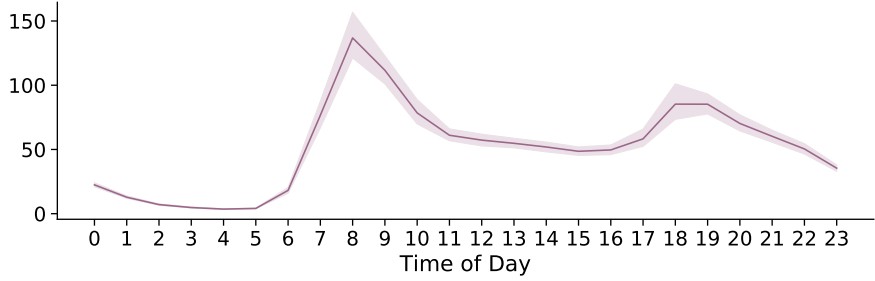

**Figure 3.** Average number of tweets per time of day with a 95% confidence interval.

With this data set, we sought to answer our research question on understanding gender differences in transport perception. Next, we explain our methodology. It is composed of three main steps (see Figure 4 for a schematic diagram): (1) user profiling used to infer gender, (2) inference of transportation topics in the discussion, and (3) measurement of perception with gender differences.

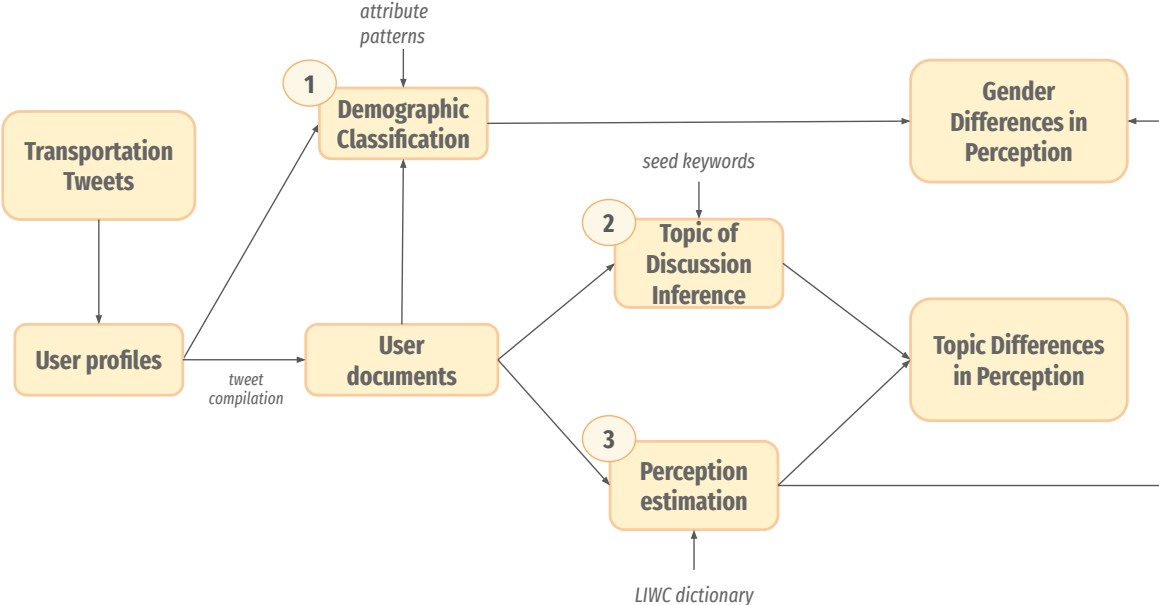

**Figure 4.** Schematic diagram of the proposed methodology.

*3.2. User Profiling*

In Twitter, user profiles contain the following attributes used in our study: an id (a unique number for each user), a full name (freely reported), a description (usually used as biography), a location (freely reported), and the number of tweets. Although Twitter requests such information, these fields are not mandatory, and can be filled with fictional data. For instance, a user could decide to write a fictional place (e.g., Wonderland) as their profile location [71], use a fantasy name, or no name at all. In this step of the methodology, we inferred gender (binary) using the available user attributes and published content by using information provided by some users—whom we could identify as residents of Santiago, with a specific gender—to predict the attributes of others.

To predict these attributes we used XGBoost [72], a gradient boosting classifier based on decision trees. Our approach was based on the idea of using profiles that self-report their attributes to train a classifier, and then to propagate attribute labels to the rest of the data set [23]. In order to classify these attributes, we characterized each user through a set of features, which included:

1. The content published in his/her/their tweets, separated into terms, where a term may be a word, a hashtag, an emoji, an URL, or an *n*-gram (i.e., *n* single terms that appear consecutively). Not all terms appear on the feature matrix, as there were many terms that were not relevant for the task at hand, such as lexical errors. Hence, we discarded terms that were used by fewer than 50 users (determined through manual exploration).
2. The content published in his/her/their biographies, separated into terms (in the same way as tweet content). Here, we use a minimum threshold of five users for each term (determined through manual exploration).
3. The profile information, which includes time-zone, the home page link, and the use of specific words related to commonly discussed topics (sports, religion, TV, etc.).
4. The interactions with other profiles through replies, retweets, and quoting other tweets.

Using these features for users, a classifier may learn that specific language cues, interactions, words in profile description, among other characteristics, have predictive power.

Next, we identified which users self-reported demographic attributes in their profiles. Gender may be revealed by displaying a typically gendered name in the profile [73], and location may be disclosed by mentioning where a user lives in the profile [71]. The set of users that we could label

though these simple measures were the training sets for the XGBoost classifiers, one for gender and one for location. Note that we used the location classifier just to discard users from the data set that belonged to non-relevant locations in the study.

Once a classifier was trained with profiles that self-disclosed their attributes, we proceeded to predict those attributes for the rest of the dataset, enabling us to have demographic attributes for all users. Note that XGBoost included prediction confidence (where 1 is maximum confidence), and thus, we only kept users with a minimum confidence of 0.7 (determined by manual exploration). Technical details about this classification process can be found in Ref. [74].

### 3.3. Topics in Transport Modeling

Some users may comment on their daily experiences, positive or negative. Other users participate in the political discussion of transport, giving their opinion on the subject. This work will focus on classifying transport discussion into two main topics: personal experience and general discussion.

Given that the text from Twitter is unstructured, the task at hand required a method to infer the latent structure that separates both aspects of content. We resort to topic modeling methods to achieve that aim. Particularly, we used Topic-Supervised Non-Negative Matrix Factorization (TS-NMF) [16], which has been used successfully in our previous work to infer the main mode of transportation of Twitter users [41], as well as mobile phone users [17].

Topic modeling techniques require a parameter with the number of topics, i.e., the number of latent dimensions in the structure inferred by the model. In general, because these topics are latent, they do not always align with human interpretation. The TS-NMF method is semi-supervised, which means that each latent dimension can be guided toward a specific meaning. A set of terms can be pre-labeled with their association to one or more topics, two in our case study. Then, the model guides the topic extraction process by aligning the obtained latent topics with the input information.

TS-NMF works by decomposing a known matrix $D$ into the multiplication of two lower-rank matrices $W$ and $H$ of rank $k$. The supervision matrix $L$ contains the available labels for user-documents (i.e., rows of $X$). The $W$ and $H$ matrices are found by solving the following optimization problem:

$$\min_{W \geq 0, H \geq 0} ||D - (W \circ L)H||_F^2, \tag{1}$$

where $|| \cdot ||$ is the Frobenius norm, $\circ$ is the Haddamard product operator, $k$ represents the number of topics, $W$ represents the relevance of each topic in each document, and $H$ represents the relevance of each term in each topic. The matrix $L$ contains the subset of terms categorized into a topic, where each row is a term, and each column is a topic, and a cell value of 1 indicates association to the topic. (An efficient Python implementation of the TS-NMF method is available at https://github.com/Vokturz/tsnmf-sparse).

In our context, $k = 2$, and $D$ is a term-user matrix, where each cell contains the weighted number of times the corresponding user published a tweet with the corresponding term. The weight is estimated using Term-Frequency–Inverse-Document-Frequency (TFIDF) [75], which measures how important a word is to a document in a collection or corpus. We apply TFIDF to $D$ to give more weight to words that are less frequent in the data set, but that may be relevant for each discussion topic. The matrix $L$ contains the subset of terms categorized into personal experience and general discussion. See Figure 5 for an overview of this approach.

After applying the TS-NMF method, we obtained two matrices, one with the association of users with the topics of discussion, and one with the association of terms with the topics of discussion. We assigned each user to a specific discussion group by estimating the maximum topic association. Users with a max topic score lower than 0.8 were discarded from the analysis due to being hard to classify (the threshold was determined via manual exploration).

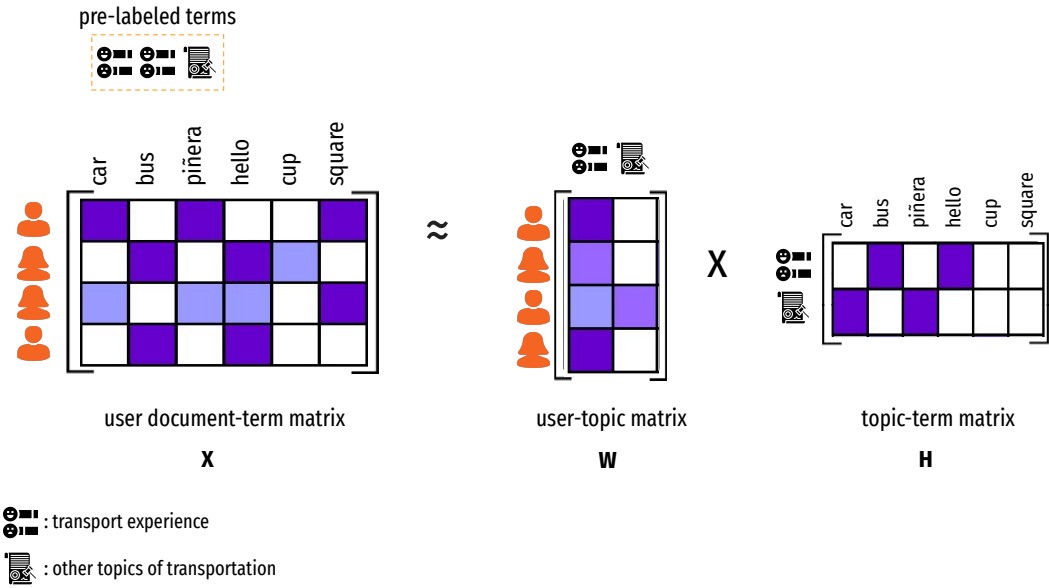

**Figure 5.** Depiction of the Topic-Supervised Non-Negative Matrix Factorization technique.

At this stage of the methodology, for each user we knew his/her/their gender and main topic of discussion.

### 3.4. Transport Perception Metric

Once users had demographic attributes and the discussion was divided in two topics, the next step was to measure perception in each discussion. To do so, we followed a lexical approach, because language style can be an indicator of the subjective way in which people understand themselves and the different situations they take part in. Particularly, we used a well-known dictionary approach based on the psycho-linguistic dictionary named Linguistic Inquiry and Word Count (LIWC) [27], which is designed to capture emotional, cognitive, and structural components present in speech. LIWC defines several categories and words belonging to these categories. The frequency of word usage is assumed to be a proxy of how these categories are related to who is performing the speech. As relevant categories for our study, we considered affective processes such as anxiety, anger, and positive feelings; relativity notions of time and space; and usage of certain words such as cursing, or sexual language (see Table 3 for examples of words from these categories). For this project, we updated the original dictionary to include Chilean slang, e.g., weón (dude, in swearing), agarrón (grab, in sexual language), and taco (traffic jam, in space).

**Table 3.** Linguistic Inquiry and Word Count (LIWC) term samples. The terms of the LIWC categories reflect cognitive processes such as emotions, and aspects related to transport.

| Category | Sample Terms |
|---|---|
| Anger | damn, assaulted, run over, aggressiveness, shit, threaten, motherfucker |
| Anxiety | riots, sorrows, scandalous, miserable, insecurity, worry, intimidation, dangerous |
| Positive Feelings | care, good, smile, want, applaud, happy, incentive, humor, wish, admire |
| Time | stopped, start, semester, minute, tip, june, late, season, we finish, start |
| Space | block, blocks, small, interior, region, zones, moving forward, base, through |
| Sexual | sexually, ass, sexual, naked, butt, darlings, fags, grabbed, rape |
| Swear | crest, assholes, faggot, asshole, fuck, shit, damn, motherfucker |

We refer to our selected categories as perception categories, because they are proxies of how people perceive the transportation experience and the general discussion. The perception of users is represented in a Perception Matrix $P$ using the LIWC dictionary, such that:

$$P(i,c) = \frac{\text{\# of times user i has used words from category c}}{\text{\# of words used by user i in any LIWC category}}, \qquad (2)$$

where $c$ is the perception category of interest, $i$ is a user id. In this way, $P$ contains emotional and contextual cues of how users feel regarding transportation, according to the frequency of words in perception categories. This matrix is divided into subgroups for the next step. For instance, two sub-matrices can be defined according to the most associated topic of users, $P_T$ (subset of $P$ with users associated with personal experience) and $P_G$ (subset of $P$ with users associated to general discussion). Four sub-matrices can be defined according to that definition, this time separating each group by gender.

Following previous work on perception in Twitter [76], we define the Transport Perception Metric ($TP$) of user $i$ in sub-group $S$ (e.g., $S$ is one of $P_T$ or $P_G$) for category $c$ as:

$$TP(i,c) = \frac{S(i,c) - \mu(S_c)}{\sigma(S_c)}, \qquad (3)$$

where $\mu(S_c)$ is the mean of column vector $S_c$, and $sigma(S_c)$ is the standard deviation of column vector $S_c$. In summary, $TP$ is a standardized measure of how each user in a subgroup expresses each category in relation to the rest of users in the same group.

Considering the previous definition of $TP$, we define the gender gap in perception as the difference in $TP$ in a category $c$ for a given discussion group $S$ as:

$$GAP(S,c) = TP(S_f,c) - TP(S_m,c), \qquad (4)$$

where $S_f$ is the group of women users in $S$, and $S_m$ is the group of men users in $S$. As result, GAP tells us if the tendency of using a specific perception category in a group is more skewed towards women (GAP > 0) or men (GAP < 0). The GAP metric allows us to calculate the gender differences in each perception category, it provides answer to our research question.

In summary, by following this methodology, it was possible to infer users' attributes, identify topics of discussion in transportation, estimate their perception in the discussion, and measure the gender differences of these perceptions. In the next section, we applied this methodology to tweets describing transportation in Santiago.

## 4. Results

In this section, we present the results of a case study of transportation tweets in Spanish from Santiago, Chile. The main goal is to characterize the gender differences in transport perception as seen on Twitter.

### 4.1. Demographic Classification

First, we report the results on demographic classification. To assess the prediction of gender, we evaluated the classifier with 5-fold cross-validation. The average precision is 0.712, a value that is below the state of the art of 0.967 [77]. To maintain the quality of the results, we only considered users that were classified with confidence. Note that our classifier relied on self-reported demographic found in users profiles, while the state of the art classifier had a more vast training set than ours, including the profile image of each user. We manually explored the dataset and found no relevant errors in gender prediction. Gender was determined by the usage of gendered-words in the biography, such as mom, engineer (man), and lawyer (woman); use of emoji, and the mention of words belonging to causes and sports (see Table 4). We classified 35% of users as women, and 65% as men. Comparing with

the Chilean population, we found that 11 women and 20 men were present in our dataset for every 1000 inhabitants. Two-thirds of users talking about transportation were men. The over-representation mirrors previous studies of demographic on the social platform [78,79].

**Table 4.** Gender classification metrics: precision and recall, and the top predictive features identified by the classifier.

| | Precision | Recall | Top Predictors |
|---|---|---|---|
| **Gender Classification** | 0.712 ± 0.02 | 0.706 ± 0.02 | profile:# of emojis, profile:mother (*madre*), profile:mom (*mamá*), profile:engineer (man, *ingeniero*), [words regarding political causes], profile:lawyer (man, *abogado*), [words about sports], profile:in love (woman, *enamorada*), profile:woman (*mujer*), profile:teacher (woman, *profesora*), profile:teacher (man, *profesor*), profile:chilean (man, *chileno*), profile:lawyer (woman, *abogada*), profile:crazy (woman, *loca*), profile:engineer (woman, *ingeniera*) |

### 4.2. Topics in Transportation Discussion

Next, we proceeded to infer the topic structure. As the first step, we built the labeling matrix *L* using a set of manually crafted seed terms. See example terms for each topic in Table 5.

**Table 5.** Sample keywords used to guide the topic modeling of transportation discussion.

| Topic | Sample Keywords |
|---|---|
| Personal Experience | bus, subway (*metro*), #line5 (#*linea5*), 🚌, @metbus, bicycle, @bikesantiago, #ipedal, 🚲, highway (*autopista*), taxi, wazers (*community of Waze users*), #vespuciosur (a highway in Santiago), cabify (a ride-hailing application) |
| General Discussion | #census (#*censo*, population census in 2017), #chileelections, (#*eleccioneschile*, national elections), #chilevotes (#*chilevota*, national elections), @maratonsantiago (a sports event) #bienvenidos13 (a TV show), #lacooperativa (news media hashtag), @elmercurio_cl (news media), @adnradiochile (news media), @mop_chile (Ministry of Public Infrastructure), @mtt_chile (Ministry of Transportation), @paolatapiasalas (Ministry of Transportation), @marcoporchile (politician), @danieljadue (politician), @camaradiputados (Chamber of Deputies) |

Regarding personal experience, we assumed that people would mention specific terms such as station names, transport operators support accounts, highway names, ride-share apps like Uber or Cabify, and terms related to cycling and walking (see Quotes 4 and 5 from Table 2). For the general transportation discussion, we expected to identify users that discuss specific events, comment on publications from media outlets, send tweets directed to politicians, or comment about transportation policy, among other themes (see Quotes 6 and 7 from Table 2 for sample tweets).

To assess the accuracy of the topic modeling method, we manually labeled a set of 389 users classified under each topic (see Table 6 for performance metrics). The precision in personal experience was 0.93 (with recall of 0.64), whereas the precision in general discussion was 0.40 (with recall of 0.83). The experience topic was correctly predicted; however, there were personal experience users that were being classified into general discussion. This was not bad per se, as our focus was the transport experience. To qualitatively explore how the topic extraction worked, we calculated a tendency value for each term in the dataset, defined as the difference between topic associations for each term. Figure 6 shows word clouds of terms with the highest tendency for each topic. Personal experience terms belong mostly to public transportation (subway—metro, line—línea–, @metrodesantiago) and often also conveyed part of the interactions between service providers and users (see Quote 8 from Table 2). They often referred to the functioning and availability of the system, with words such as working (funciona), available (disponible), problems (problemas), service (servicio), and thank you

(gracias). Meanwhile, terms related to general discussion consisted mostly of mentions of politicians and service administrators (see Quote 9 from Table 2), as well as media outlets (@biobio) and politicians (@sebastianpinera, president of Chile).

**Table 6.** Manual evaluation of the topic model.

| Topic | Precision | Recall | F1-Score |
|---|---|---|---|
| Personal Experience | 0.93 | 0.64 | 0.76 |
| General Discussion | 0.40 | 0.83 | 0.54 |

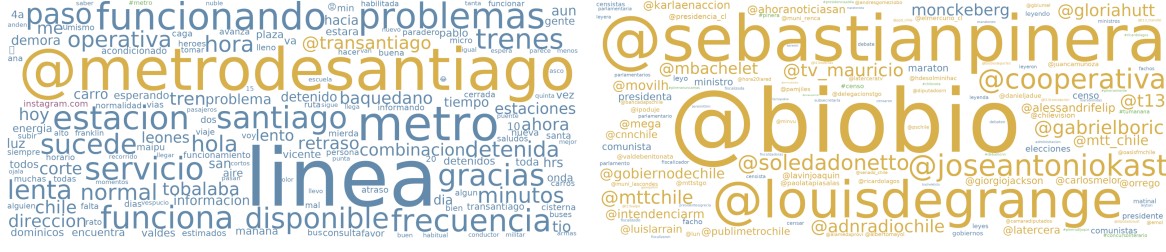

(**a**) Personal Experience                          (**b**) General Discussion

**Figure 6.** Word clouds of the most associated terms to each topic of transportation.

The frequency of tweeting among topics showed resemblance to the general tweeting frequency, where transport discussion was higher during the weekdays to decrease over the weekend (see Figure 7). The relationship between both topics, in relative terms, was maintained during the whole week.

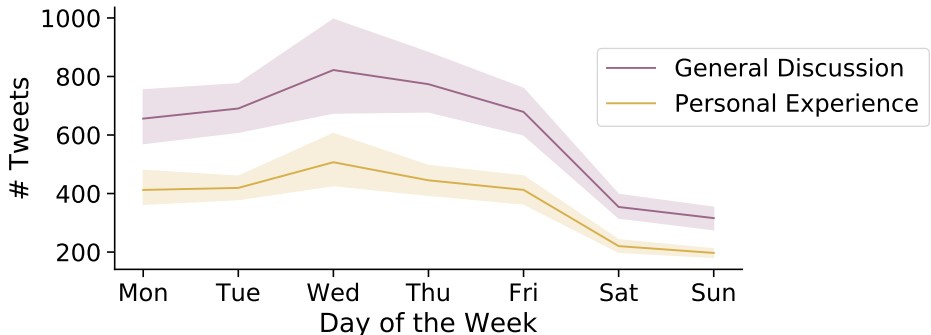

**Figure 7.** Tweet frequency per day of week, by topic of interest.

With respect to gender, a difference appeared in the distribution per topic. In general, men represented a larger portion of both topics, and participated more in the general discussion, which means that men used Twitter as a platform to express their opinions on politics, event discussion, among others. We expected a majority of Twitter users to be male due to the relative demographics of the population in Chile, and therefore even topics more likely to be gendered towards women have an greater representation by male users [80]. However, in proportion, women appeared to talk more about their daily experiences when referring to transportation (see Figure 8).

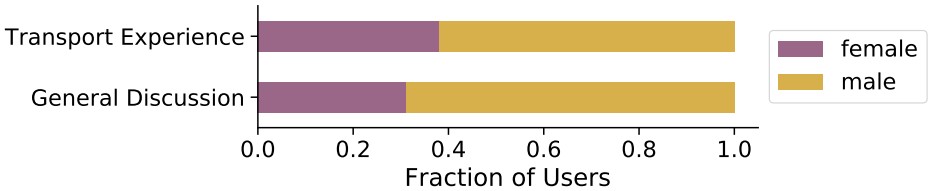

**Figure 8.** Distribution of users per topic of transportation and gender.

### 4.3. Transport Perception

Following our methodology, the next step was estimating transport perception using LIWC categories in each discussion topic. As an exploratory analysis, we estimated the correlations between each category in the perception of users as defined in matrix *P* (see correlation matrices in Figure 9). The correlation between two different categories could help us understand the interactions between each of them. Time and space had a negative correlation in both personal experience and the general discussion of transport. This may be due to users only referring to one relative component in general, probably the one with the greatest personal importance. In that sense, time and positive feelings correlated negatively, which implies that users did not often express themselves positively while referring to travel times (see Quote 10 from Table 2). On the other hand, the general transport discussion had its focus on the space-related component. This can be caused by users commenting on the design or planning of transport, commenting on vehicle congestion, or the little space inside the train or bus cars, among other things (see Quote 11 from Table 2).

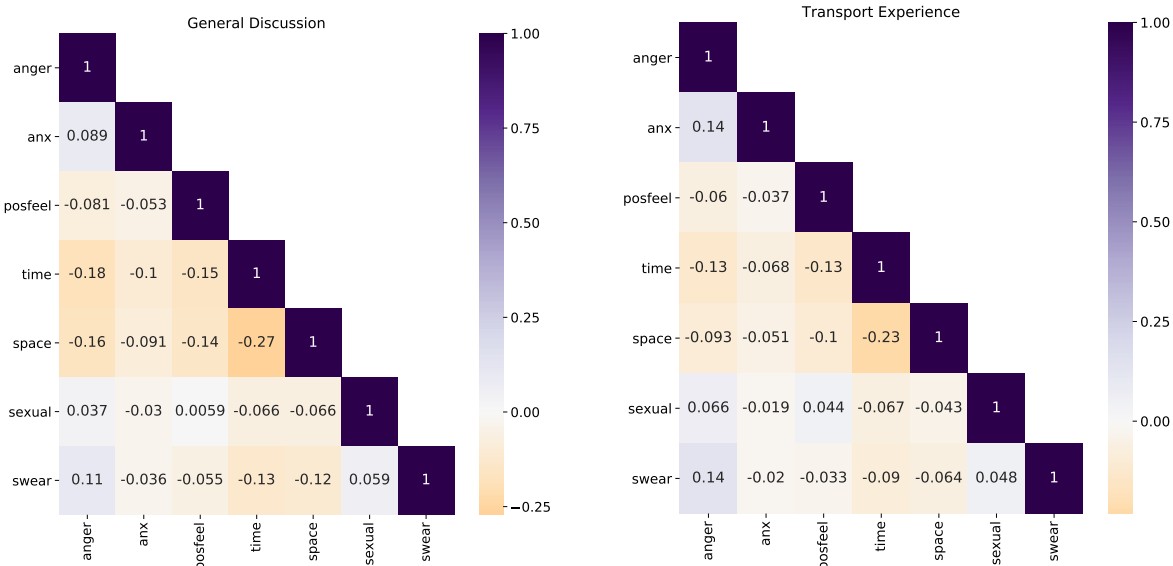

**Figure 9.** Matrix of significant correlations between Perception categories for the Personal Experience topic. All correlations are significant ($p < 0.05$, corrected using the Holm–Sidák method)

Regarding emotional components, the use of words related to anxiety and anger were positively correlated in the personal experience topic, which was an expected result when people experience uncertainty in their travel. Positive feelings were negatively correlated with the space component for the personal experience discussion, perhaps related to the discontent users experience when they do not have enough space. Swear words were positively correlated with anger for both topics, an arguably expected result.

Then, we proceeded to estimate the differences with respect to gender in the overall discussion (see Figure 10). We found that women showed greater use of positive and anxious words, while men used more swear words. Other insights also emerged, with women using more words related to time, and words from the sexual category (See Quotes 12, 13, 14, and 15 from Table 2). Evidence showed that time played an important role in the experience of women and their interaction with transportation. There was also greater use of words in the sexual category depending on the time. These results are consistent with previous results on the issue [81].

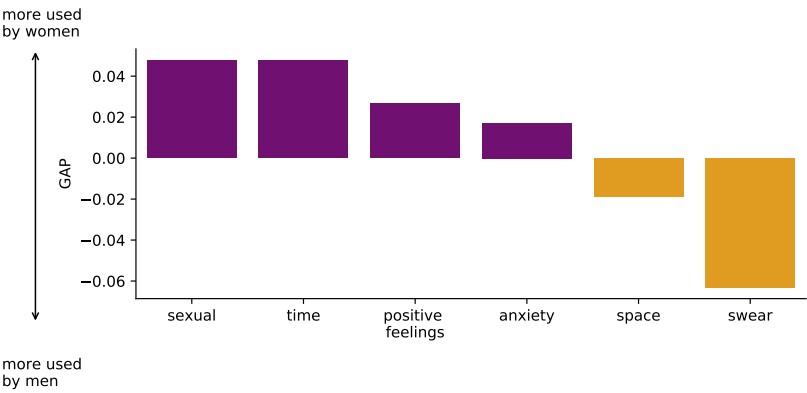

**Figure 10.** Gender gap in perception.

Finally, we estimated the differences in perception by gender for the two topics of discussion. Women expressed anxiety and positive feelings more frequently than men, in both dimensions of transport, skewing more to time. They also use sexual-related words, likely participating in the discussion about harassment. This indicates that while harassment was something they experienced on a daily basis, it was seen as a systemic problem within transportation, which was also discussed from the approach of politics and planning. There were no differences in anger on the general discussion, but women expressed more anger in reference to their transportation experience, implying that there were aspects of transport where users did not feel satisfied. In addition, women used more words related to time when talking about their personal experience, which indicated the relevance of this factor. We validated these results following a null model approach, where we estimated 1K tests shuffling the gender attribute of profiles. Figure 11 shows the comparison between the distribution of observed values (Effect Model, with distribution estimated using bootstrap sampling with replacement) and the null model.

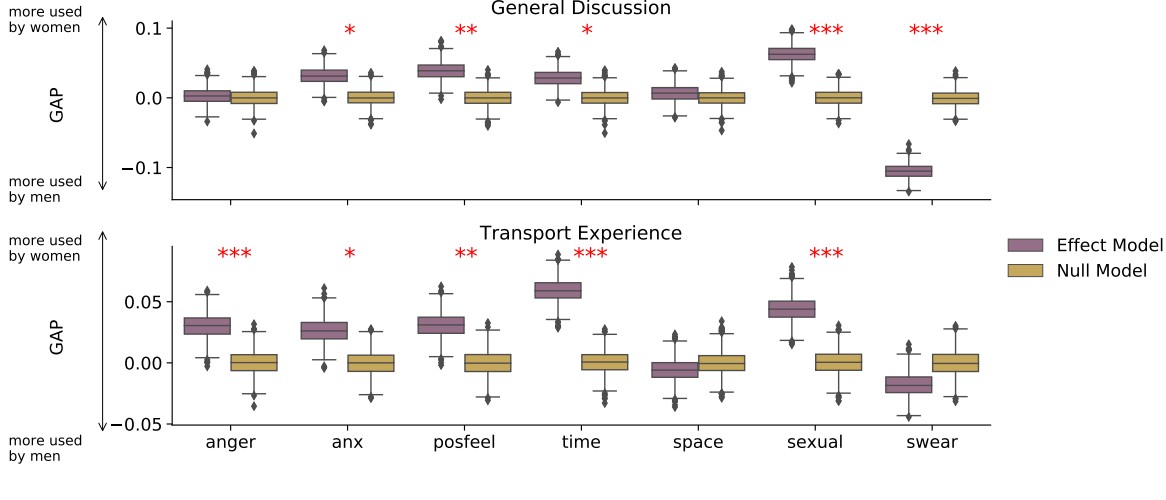

**Figure 11.** Boxplots of the GAP distributions per gender, topic, and perception category. Each plot contains the observed distribution (estimated with bootstrap sampling) and the null model (estimated with feature permutations).

## 5. Discussion

The main aim of this paper was to design a methodology to measure the gender differences in transport perception as seen on social media platforms. We measured the gender differences across seven perception categories: affective processes (anxiety, anger, and positive feelings), relativity notions

(such as time and space), and the use of words associated with cursing or sexual issues. In particular, we conducted a case study of gender differences in perception in micro-blogging messages from Santiago, Chile. Here we showed that both genders express differently. On the one hand, female users write about their emotions regarding travel (both positive and negative feelings). On the other hand, male users use more slang, making it difficult to interpret emotion. The most notable difference is related to female sexual-related harassment, not only on transportation, but also on the public space. This may indicate that women are more exposed to sexual harassment in transportation, a situation reported in other big cities in Latin America [14].

Overall, we believe that data obtained from social media can complement traditional information gathering methods, overcoming restrictions such as time and cost, while also allowing for previously unseen granularity of data. In this context, here we discuss the implications, limitations, and future work derived from this paper.

### 5.1. Comparison with Previous Works

A previous study on gender differences in language use has shown that women often tend to express anxiety or positive feelings more frequently, with no differences in anger word usage [81]. However, for the personal experience discussion, women showed significantly more anger in their publications, proving their dissatisfaction with their journey experience. At the same time, women also referred more to time while talking about their personal experiences, which is consistent with previous studies on the perception of quality factors in which women place a high emphasis on punctuality in transportation systems [59]. The most significant gender difference is related to harassment to women, not only on transportation but also on the public space. Both issues are consistent with recent studies in Latin American cities [82].

Our results contradict previous assessments of sentiment toward public transport in Santiago de Chile, where it was found that most messages regarding transport experience were negative [69]. From a dataset point of view, both approaches are not directly comparable, as the referenced study focused on tweets about bus stops and bus routes, which may prompt adverse reactions when published on the go. From a methodological point of view, the study did not consider differences concerning gender, which may influence results considering the over-representation of men in Twitter [21,80]. Addressing this disagreement could be done by defining more specific topics in future work.

### 5.2. Implications

Social media analysis enables the identification of significant gender differences in the transport experience. Historically, these differences have been hidden due to the consideration of the "average user" in transportation. This identification of differences may be helpful for service providers and policymakers, not only in terms of management and planning, but also to improve accessibility of the public transportation with a gender perspective, in line with the Sustainable Development Goals by the UN, which aim to make cities more inclusive for all.

Service providers could track the most used terms to refer to transportation on their social networks, and study their occurrence based on events, time or characteristics of their users. Our method would allow them to highlight particularities that a survey or a traditional method fails to show.

Another benefit of having gender indicators is that it makes it possible to monitor the situation of women in transport. It allows evidence-based decisions to be made by policymakers to design, formulate, monitor, and evaluate interventions effectively. We focused our analysis on Santiago, Chile, however, harassment in the public space and transportation is a common problem in Latin America, thus, our work could be generalized and applied to other cities such as Quito, Buenos Aires, among others [82].

*5.3. Scope and Limitations*

Critics may point out three aspects that scope the reach of this work: representativeness, geographical coverage, and methodological points of improvement.

First, we measured and compared transportation discussion with respect to gender, however, there are also other population groups with special needs, such as children and the elderly. Including them in this kind of study is not just a matter of adding categories to the demographic classifier, as these groups are also under-represented in social media in general. This aspect does not hinder the insights obtained: as long as it is made clear who is represented by the study, then insights are valuable and actionable.

Second, the lack of geographic information prevents us from comparing different sectors of the city, which may be crucial for transportation discussion. For instance, comparing the wealthier part of the city with deprived sectors may reveal different emotional patterns with respect to the transportation experience.

Third, our methodology can be deepened in several steps. For instance, users are assigned to their most associated topic, whereas a more advanced alternative would be to assign weights to topics. The consideration of the temporal aspect of discussions is also needed, as it would allow answering questions such as "what is the time of the day perceived as more safe/dangerous for women?" Still, our work has proven that it is possible to move in this direction. We leave these advancements to future work.

*5.4. Future Work*

In addition to address the limitations of this work, we devised two main lines of research: visualization and bias mitigation. Our current results provide insights regarding the transportation discussion, with identified implications for two types of stakeholders. However, it is not clear how to actually transfer those insights to these stakeholders in a meaningful way. Visualization has been a successful medium to communicate between data scientists and transportation experts, particularly in scenarios of solving transportation problems with non-traditional data sources and methods [83]. Finally, social media data is not considered a representative source of population information [21]. Several access gaps to technology, plus the different algorithmic and systemic biases in operation in these scenarios, imply that designing and implementing bias mitigation strategies are needed, especially if the insights derived are planned to be used in decision making and policy design.

## 6. Conclusions

In this paper we quantified the gender differences in perception of transportation from a two-fold perspective: the personal experience of transport, and the general discussion around it. The usage of social media data, particularly from Twitter, which is not used traditionally for these issues, analyzed with methods from machine learning and psycho-linguistics, proved fruitful in finding topic- and gender-specific insights.

If these differences are taken into account by relevant stakeholders in city planning and management, policies could be implemented that help eliminates gender gaps in terms of access to work, education, opportunities, and above all, equal access to the opportunities a city has to offer, in line with the Sustainable Development Goals. Future work should address methodological aspects of our proposal, but also communication and transfer strategies to ensure its adoption in planning, complementing survey-based information, making visible critical problems that are hidden in these traditional instruments.

**Author Contributions:** Conceptualization, P.V.-H. and E.G.-G.; methodology, P.V.-H. and E.G.-G.; software, P.V.-H. and E.G.-G.; validation, P.V.-H.; formal analysis, P.V.-H. and E.G.-G.; investigation, P.V.-H. and E.G.-G.; resources, E.G.-G.; data curation, E.G.-G.; writing—original draft preparation, P.V.-H. and E.G.-G.; writing—review and editing, P.V.-H., E.G.-G., and D.C.; visualization, P.V.-H.; supervision, E.G.-G. and D.C.; project administration,

P.V.-H. and E.G.-G.; funding acquisition, E.G.-G. and D.C. All authors have read and agreed to the published version of the manuscript.

**Funding:** P.V.-H. and E.G.-G. were partially funded by CONICYT Fondecyt de Iniciación project #11180913. P.V.-H. and D.C. were partially funded by Concurso Interno de Investigación UDD #CI18.

**Acknowledgments:** The authors thank the anonymous reviewers for their helpful comments in improving this paper.

**Conflicts of Interest:** The authors declare no conflict of interest. The funders had no role in the design of the study; in the collection, analyses, or interpretation of data; in the writing of the manuscript, or in the decision to publish the results.

## Abbreviations

The following abbreviations are used in this manuscript:

| | |
|---|---|
| API | Application Programming Interface |
| GAP | Gender Gap in Perception |
| LDA | Latent Dirichlet Allocation |
| LIWC | Linguistic Inquiry and Word Count |
| NMF | Non-negative Matrix Factorization |
| SMA | Social Media Analysis |
| STS | Satisfaction with Travel Scale |
| TFIDF | Term Frequency–Inverse Document Frequency |
| TP | Transport Perception |
| TS-NMF | Topic-Supervised Non-negative Matrix Factorization |
| URL | Uniform Resource Locator |
| XGBoost | Extreme Gradient Boosting |

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
