# Peer review of "Tweets on the Go: Gender Differences in Transport Perception and Its Discussion on Social Media"

_sustainability, doi:10.3390/su12135405_

Round 1

Reviewer 1 Report

see the file attached

the authors add to the topic of gender in trasnportation by including the possibility of inferring information from social media tweets related to travel experience and general discussion on transportation topics.

i enjoyed reading the paper; nevertheless some improvement is still needed. mainly i would underline three topics:

  • i suggest a few more literature on the gender & transportation theme and travel perception; 
  • i recommend language editing and revision. i attempted to re-formulate some sentences within the paper. I hope to be helpful
  • a revision of the relative position of figures/tables within the paper is advisable, as often figures are placed many lines before being called by the text. This is not helpful as - while reading the paper - the reader is forced to scroll up/down to locate the figure to which reference is made.

Author Response

Dear Reviewer,

Thank you for your constructive comments. Please see the attachment for the point-by-point revisions.

Reviewer 2 Report

Thank you for submitting this very interesting article to the journal. The article is interesting, but there are some concerns that should be fixed, if possible, before further processing:

  1. the paper is not really on the topic of the journal. The authors should try to find a more suitable way to connect there very interesting paper with the topic of sustainability.
  2. The research question of the paper is not really good stresses out. It should be more clear what the main aspect that the authors investigate is and how the paper enhances the literature.
  3. A proof read of the paper would be useful, as there are different little aspects that need to be corrected

The major concern:

In my opinion there are 2 papers in this paper. The authors should try to focus on one idea and not on two. Either they focus on the methodology and explain the scope of the new methodology or the apply the new methodology. But making both ... is really hard to follow the paper.

Other aspects to consider

Title: "Transportation Discussion " which discussion? Discussion among who? A title should be short and clear and it should reflect the content of the paper. Your present title is not very "sexy" / interesting. Academic writing papers recommend to have a "triade" title-abstract-key words. These 3 sections are not really consistent. 

Santiago, Chile is the research context. I would not state in the abstract that you refer to Santiago. Maybe you could speak of an emerging market and afterwards explain why you choose that particular context: Santiago and why not another one.   

Abstract: please be more concise regarding the research question and how the paper is based on a theory / enhances the literature.

"routes and behaviour such as destination" ... destinations

"o their publications " well I do not think that these are publications. Probably you want to speak of opinions ...

Introduction

General comments:

  • one phrase / sentence SHOULD never be a paragraph of its own. One paragraph consists of several phrases.
  • A research paper should have / deal with only ONE research question (RQ), not with several ones. I advice you to reconsider your research questions. Of course, a research question can be divided into one or several research objectives
  • You need to start with the research gap identified in the literature and explain how this gap is transposed into the RQ.  
  • a proper introduction should have about one page
  • figures in the introduction are not the best choice. 

Usually a figure should be introduced first in the text. So putting the fig 1 on page 2 and the text on page 3 is quite... strange.

L 75-80: this information is probably from some public statistics. I would use that when presenting the research context. In the intro you only need to specify some minor aspects that can justify your research context.

The introduction needs to be rewritten and shortened. Speak of the research gap, the research question, the theory behind the paper and explain how the research scope is related to an existing theory and how it enhances the literature. Explain briefly the research context and also how the RQ is implemented in the different sections of the paper, especially in the methodology you employ.

The last paragraph of the introduction should present the next sections of your paper.

You need to make the discussion around the "transportation discussion"more coherent. 

The paper does not have a literature review section. I therefore recommend the authors to extent the paper with such a section

It seems that your choice of the research context is Santiago and Twitter. Why this 2? You need to speak about them in the section "research context" not in the lit review section.

This paper seems not to have a proper lit review. I would recommend also inserting a lit review section.

Table 1 and 2: do you have the right of publishing this data?

L 117-120: this is more research methodology. This is not literature review. 

So ... your paper is about proposing a methodology for ... ? figure 5. This should be clear from the beginning of the paper. What is the purpose of this new methodology? How does this new methodology enhance the literature or the practice in the field?

The paper is very interesting, but it is very hard to follow. It does not really have a clear and concise focus and a kind of "red line". It seems that there are kind of 2 papers and 2 methodologies. One is for the tweets and one is for the case study. If you propose a new methodology, than the entire focus of the paper should be another one. There are other papers which propose new methodologies (in other context). You should try to find them and do as they have done before. The information is now kind of mixed up and difficult to follow.

Discussion: if you have a methodology... than I do not really see the necessity of the discussion. BUT it would be proper to try to apply the new methodology that you propose.

Related work: I do not understand the logic of this section here. This is kind of literature review? Then it should be at the front of the paper.

Conclusions

What are the theoretical implications?  Are there limitations?

The paper needs to be restructured and rewritten.

Author Response

(The authors gave the same response as above.)

Reviewer 3 Report

Dear Authors,

the paper is very interesting, consistent with the latest research in the field of communication behavior. It is worth paying attention to its very good structuring, the individual parts are logically described and connected with each other. The research presented in the article is supported by examples from a comprehensive literature review. The conclusions are detailed and reflect the content of the article well. In fact, I have one remark regarding the Figure 1 presented in the introduction. I could not find the text in the lead to it. I propose to complete this and indicate in the text exactly which part it concerns. 

Author Response

(The authors gave the same response as above.)

Reviewer 4 Report

With great interest and pleasure I read an article devoted to the study and measurement of gender differences in transportation. The use of social media data seems to be a particularly interesting solution - both in terms of methodological innovation and access to big data, the use of which in scientific research has still not reached its full potential. Extremely interesting conclusions find not only empirical applications, but allow the development of similar research in the future. Profiling, which combines demographic and psychographic features, seems to be particularly important in the area of social sciences. The authors show how interesting effects the dynamic approach of certain phenomena can bring, combined with the relatively low cost of the study itself, which is a response to the numerous challenges faced by researchers.

Author Response

Dear reviewer,

We are thankful for your kind comments on our two-year’s work. We are happy you enjoyed our paper.

Reviewer 5 Report

The paper focuses on measuring how the different aspects of the travel experience are perceived by men and woman, using as a data source the the social media messages posted on Twitter.

The topic is interesting and worth investigating. Also, the study is based on a relatively large dataset, collected over a long period of time, which has the potential to provide a good overview of the travelers' perception, without being affected by specific events. 

# Recommendations

  • It would be advisable to follow the classic structure for a paper, in which a literature review section follows the introduction. In the present version of the paper, the authors mention other approaches using social media analysis for the transport domain just before the conclusions section.
  • The literature review section should also be extended to better mention existing studies that use social media analysis in the transportation domain. Currently, other approaches are only briefly mentioned. 
  • Several gender classification approaches in the context of social media analysis currently exist. The authors are kindly asked to mention them and to compare their approach with existing ones in terms of precision / recall.
  • The authors are kindly asked to explain their choice for the frequency thresholds of 50 for terms in D, and 5 for terms in B (line 162).
  • The language in which the analyzed tweets have been written should also be clearly specified.
  • In the Introduction section, the paper mentions the research questions as RQ1 and RQ2. The same name should be used in the Discussion section, when commenting the results.

# Suggestions

  • It would be recommended for the authors to share the original data set of tweets in order to facilitate the reproducibility of the results. 

Author Response

Dear Reviewer,

Thank you very much for your constructive comments. Please see the attachment for the point-by-point revisions.

Round 2

Reviewer 2 Report

The paper has been improved considerably. Changes have been implemented according to my suggestions. I congratulate the authors for that.

There are still some minor aspects that should be considered before further processing:

  1. Please try to have a broader discussion - namely please try to make more comparisons between your results and similar results from the literature.
  2. Please extend the conclusions with 4 paragraphs: a) theoretical implications of the research and the results; b) managerial implications of the results; c) limitations of own results; d) future research perspectives.

Please also try to strenghten the connection with this journal.

Conclusions refer to the own work. Therefore no other references should be included here!

Implications and limitations SHOULD always be part of the conclusions. Discussions should contain a comparison between own results and similar results in the international literature!

Author Response

Dear reviewer,

Thank you for your relevant suggestions. Please see the attachment for the point-by-point response.

Reviewer 5 Report

I would like to thank the authors for thoroughly addressing the comments in the previous review.

Author Response

Dear reviewer,

We are thankful for your previous suggestions. They have been very helpful in improving the manuscript. We have acknowledged this in the paper (lines 481-482).